# Evaluating the Performance of Airborne and Ground Sensors for Applications in Precision Agriculture: Enhancing the Postprocessing State-of-the-Art Algorithm

**DOI:** 10.3390/s22197693

**Published:** 2022-10-10

**Authors:** Karel Pavelka, Paulina Raeva, Karel Pavelka

**Affiliations:** Department of Geomatics, Faculty of Civil Engineering, Czech Technical University in Prague, 166 36 Prague, Czech Republic

**Keywords:** NDVI, sensors, agriculture, python, gdal, pyQGIS

## Abstract

The main goals of the following paper are to evaluate the performance of two multispectral airborne sensors and compare their image data with in situ spectral measurements. Moreover, the authors aim to present an enhanced workflow for processing multitemporal image data using both commercial and open-source solutions. The research was provoked by the need for a relevant comparison between airborne and ground sensors for vegetation analysis and monitoring. The research team used an eBee fixed-wing platform and the multiSPEC 4c and Sequoia sensors. The authors carried out field measurements using a handheld spectrometer by Trimble—GreenSeeker. There were two flight campaigns which took place near the village of Tuhan in the Czech Republic. The results from the first campaign were discouraging, showing less possibility in the correlation between the aerial and field data. The second campaign resulted in a very high percentage of correlation between both types of data. The researchers present the image processing steps and their enhanced photogrammetric workflow for multitemporal data which helps experts and nonprofessionals to reduce their processing time.

## 1. Introduction

Information about the status of a certain agricultural crop could be used to improve the managerial processes that take place on any farm. Image data provides us with a promising contactless tool for monitoring vast fields. Vegetation has been monitored from space since the beginning of the 1970s [1]. Nonetheless, multispectral photogrammetric mapping is demanding, and it could theoretically differ from classical photogrammetric imaging. Spectral bands other than the classic red, green, and blue bands usually have lower image resolution than RGB (red-green-blue) ones, which directly affects the planning of unmanned flights. Scene reconstruction and reflectance orthomosaics depend on many factors, but most importantly, they are dependent on the overlap between neighboring images [2].

Imagery data could help generate maps of current plant status as well as produce yield maps. The latter could show yield patterns within the monitored area. The maps could also improve field characteristics such as drainage, land levelling, irrigation, fencing, and so on [3]. Sensors and platforms have become financially accessible, which has led to a great abundance of these devices on the market. Researchers have begun comparing consumer-grade sensors to expensive ones [4,5,6].

The topic investigated here is very popular at present, with numerous studies and tests published in books and journals [7,8,9,10]. One of the most common studies is the comparison between in situ data and aerial data where the research team showed great potential correlating the two types data [5]. Another study claims that there is a higher correlation between a spectrometer and a multispectral camera rather than a modified one [4].

The great abundance of sensors and remotely piloted aircraft sometimes makes it difficult to choose the right technologies for a certain task [11]. As the available technology and market sales increase by leaps and bounds [12], today, one can find economically affordable four-band sensors combining bands with near-infrared to infrared [13]. However, multispectral mapping still possesses certain ambiguities which may lead to erroneous image postprocessing. One very popular problem as far as multispectral sensors are concerned is the inner correlation between data [14].

Remotely sensed data are crucial for agricultural analyses because they provide variable chlorophyll content, biomass, and so on [9]. However, for automatic calculations from remotely sensed data, a calibrated sensor, atmospheric effects, and bidirectional reflectance properties of the surface must be taken into account. Most sensor manufacturers provide a calibration procedure [13,15]. Despite that, recent researchers published laboratory and field test calibrations which significantly increase the one stated by the manufacturer [16,17]. Nonetheless, there are few research papers evaluating the performance of two different sensor types. Moreover, sensors differ in their performance. In order to use a near-infrared camera for multitemporal analyses, one must be acquainted with the sensor stability during different conditions.

There are many unsolved issues concerning multispectral mapping. Some of them are considered as errors and a potential threat to commercial unmanned flying [16]. For example, a firmware update might cause performance changes [18]. Another issue might be precise geolocation of the mapped object which is corrected by using ground control points providing georeferencing to the dataset. However, the most common problem is sensor performance which is guided by the radiometric calibration applied.

However, the following study focuses on evaluating the performance of two near-infrared sensors during different conditions and comparing the image values with ground-measured data. The authors compare the image values from two aerial sensors with one handheld spectrometer for ground measurements. The first sensor multiSPEC 4c is a four-band multispectral camera with a predefined calibration procedure. The second aerial sensor is the Sequoia which captures image information in RGB, red-edge, and near-infrared bands. It possesses a sunshine irradiance which is claimed to improve the calibration [19]. The manufacturer has a predefined calibration procedure as well. Both cameras are presented to be professional in the agricultural field. That is why, the authors set their goal to investigate whether they perform in a similar way under the same conditions. Moreover, one of the team targets was to compare the aerial data with in situ information.

From a user point of view, it is important to choose the most user-friendly option to monitor the current state of the crops. On the other hand, from a scientific point of view, it is necessary to investigate if these sensors perform somewhat similarly or equally. The article will give the answers to these questions.

Gathering multitemporal aerial data for any research, the authors encounter numerous problems with regards to computing vegetation indices and storing data. Therefore, an enhanced workflow is presented where commercial and open-source solutions are applied.

## 2. Materials and Methods

The following research was conducted in 2020 at the Department of Geomatics at the Czech Technical University in Prague. The authors used a fixed-wing eBee platform, two aerial sensors compatible with it, multiSPEC 4c and Sequoia, and a handheld spectrometer for ground data. The goal of the authors was to evaluate the performance of two mainstream aerial sensors, multiSPEC 4c and Sequoia, both manufactured by AIRNOV and senseFly [13,20].

The multiSPEC 4C sensor can capture images outside of the visible parts of the spectrum with central wavelengths of its four bands as follows: green 550 nm, red 660 nm, red-edge 735 nm, and near-infrared 790 nm [13]. The imagery produced by this camera is an 8-bit single-band grayscale image. The photometric interpretation is BlackIsZero, meaning the value 0 is represented as black. Band characteristics of the multiSPEC 4c are summarized in Table 1 and Table 2.

The Sequoia sensor is a five-band camera capturing images in both visible and near-infrared spectra. The output imagery is grayscale. The camera possesses a sunshine sensor and provides irradiance corrected data. Due to the latter, the Sequoia sensor is claimed to be an advanced camera created especially for agricultural mapping. The sunshine sensor has the same filter for 4 spectral sensors, GPS, inertial measurement unit, and magnetometer. The near-infrared images have a 1.2 Mpx resolution, and the RGB have 16 Mpx. Unlike the multiSPEC 4c bands, the green and near-infrared bands are narrower. The Sequoia characteristics are listed below in Table 3.

A handheld spectrometer, the GreenSeeker from Trimble [21], was also used in order to compare its results with the aerial ones. This is an optical sensor used for measuring crop biomass and directly computing NDVI values. The spectrometer is supposed to be held 60 ÷ 120 cm above a crop (see Figure 1). Its field of view has an ecliptic form that becomes larger, the higher the sensor is held. When held 60 cm above the ground, the a-axis is approximately 12 ÷ 13 cm.

The measured data are stored in the inner memory of the instrument and can be later downloaded into *.csv format. Unfortunately, the GreenSeeker does not have a GNSS receiver. For that reason, in situ data were georeferenced with a geodetic GNSS receiver. The spectrometer was kindly leased by the Czech company Leading Farmers, S.R.O., solely for the purpose of this research.

Photogrammetric flight campaigns were carefully planned in advance in order for the specified research purposed to be fulfilled. Flight plans were programmed in eMotion 3 using senseFly. A triple image overlap was insured to eliminate radiometric ambiguities when orienting the images.

During the first flight campaign on 29 May 2020, in situ measurements were carried out with the handheld spectrometer in a grid of approximately 2 m (see Figure 2). Every single NDVI ground measurement was georeferenced with a GNSS receiver. Altogether ninety-five field measurements were taken. Ground control points were set for better georeferencing between the different sets.

During the second flight campaign on 29 July 2020, flights with both aerial sensors were planned around sunrise, at noon, and before sunset. The exact timing of the unmanned flights is summarized in Table 4.

Two types of ground control points (GCP) were used to georeference the datasets. Altogether ten points were measured with five classic checkerboards and five boards wrapped in aluminum. GCPs were measured using a GNSS receiver in JTSK Krovak and later transformed into a JTSK Krovak East (epsg: 5514). The authors decided to test the stability of the handheld sensor. For that, eight specific points were measured during the campaign with the GreenSeeker. The measurements took place five times during the day of 29 July. The field points were also georeferenced with a GNSS receiver. These points had a significantly different texture than the surroundings (see Figure 3) which made them distinguishable in the aerial imagery.

The authors chose to perform photogrammetric flights around sunrise, noon, and sunset on 29 July. In that way, it would be possible to monitor the performance of the aerial sensors and estimate the influence of the sun on the resulting maps. All multispectral flights during 29 July were mapped according to the sun elevation on that day (see Figure 4). Performing numerous flights leads to generating big imagery data which sometimes might be hard to handle and process immediately.

Figure 4 shows the time schedule of all flights during 29 July 2020. On the chart of the solar elevation for the certain day, the field measurements were also charted. The correct starting times of the flights are summarized in Table 4.

## 3. Methodology

In the following chapter, the image preprocessing steps will be explained. These technological steps refer only to the aerial sensors used. Moreover, it will discuss the solution of the authors to process numerous imagery datasets. Creating the photogrammetric framework was one of the research goals.

Image preprocessing was performed with the Swiss software pix4Dmapper [22]. This software product uses the structure-from-motion method [23]. The authors claim that postprocessing in pix4Dmapper is the best option when mapping with a senseFly sensor as pix4Dmapper can read coded Exif metadata from the senseFly images that other products cannot [24]. The processing scheme follows classical steps when processing photogrammetric measurements.

Image processing is crucial, especially when processing multispectral imagery which has a lower resolution than classic RGB imagery. This process consists of extracting identical points or key points as named in the pix4Dmapper. These points are later matched with their counterparts from the overlapping images. The authors planned all flights with a triple image overlap in order for radiometric ambiguities to be eliminated. When the identical points are matched, this lays the foundation for computing the automatic aerial triangulation and bundle block adjustment [23].

When mapping with a multispectral sensor, the geometry of the object and camera is important, but one must also consider how the object reflects energy. In the initial steps of preprocessing, one is extracting the properties of the reflecting energy, which is later compared with other datasets.

In practical terms, image orientation and model reconstruction depend on matching based on identical points between images. All forms of infrared imagery which should be extracted are crucial for image alignment.

The software product that the authors chose can extract important information from the Exif metadata. The initial processing extracts parameter values such as exposure time or ISO. These are used for basic radiometric corrections.

Two aspects affect the image preprocess: dark current and vignetting. Dark current shows the dark or black current, the measured values without any light. The values for dark current are uniform for all pixels. The important metadata that an image should contain are their Exif, Xmp.Camera.BlackCurrent, as stated in the image tag. Image vignetting is a reduction in the size of the entrance pupil for off-axis objects caused by the physical properties of the lens. Usually, cameras transmit more light in the center than in the corners. Thus, vignetting could be represented by a raster map that has a center value equal to 1 and corner values smaller than 1. Vignetting can be represented in the form of a 2D polynomial or as a radial polynomial. The 2D polynomial is defined according to the formula:(1)$$\nu =\overset{N}{\underset{i,j=0}{\sum }}c_{ij}{\left(\frac{x}{w}\right)}^{i}{\left(\frac{y}{h}\right)}^{j}$$

The radial polynomial is in the form of the following equation:(2)$$\nu =1+\sum_{i=1}^{N}c_{i}r^{i},$$
where $r=\sqrt{{\left(x-c_{x}\right)}^{2}+{\left(y-c_{y}\right)}^{2}}$ and the values $\left(c_{x},c_{y}\right)$ are read from the Xmp.Camera.VignettingCenter and the coefficients read from the Xmp.Camera.VignettingPolynomial from the Exif data.

The process of geometric camera calibration consists of optimizing the internal and external parameters of the camera. Due to the low image resolution, a higher image overlap was set.

The unmanned flights were prepared in the senseFly software eMotion 3.4.0. The goal was to set a desired image resolution (10 cm) and create an overlap of 90%. This could be achieved by using a triple overlap, namely, flying from east–west, west–east, south–north, and north–south. In that way, the authors presumed that ambiguities with the radiometric calibration of the corner pixels will be eliminated.

For the postprocessing scheme, the authors chose a geometric calibration method that excluded the oblique images. This method requires at least 75% of geotagged images. An oblique image in that case means an image that has a yaw angle higher than 35°. Not using such a geometric camera calibration might lead to reconstruction obstacles [25].

A radiometric calibration procedure was carried out during each flight campaign. The radiometric panel was put horizontally on the ground with no objects casting shadows or blocking the panel. These images were later processed in the software, where raw reflectance digital numbers (DN) were computed to absolute reflectance values. The purpose of this operation was to quantify incoming radiation (irradiance) dependent on the measurement site. The spectral reflectance of the object was computed by rationing the reflected energy measurement in each band. The result of such computation is often called the reflectance factor. This factor is defined as the ratio of the radiant flux reflected by a sample surface to that which would be reflected into the same geometry by an ideal surface (Lambertian) in the same way.

Another possible measurement of the reflectance factor is when one direction is associated with the sample viewing angle (0°), and the other direction is that of the illumination of the sun (defined by the solar zenith and azimuth angles).

The amount of reflected energy can be characterized by the spectral reflectance ρ(λ) defined as at the ratio between the intensity of the incident radiation *Mi* and the intensity of the reflected radiation *Mr*.
(3)$$\rho \left(\lambda \right)=\frac{M_{r}\left(\rho \right)}{M_{i}\left(\rho \right)}\cdot 100\left[\%\right]$$

The reflectance values provided by the manufacturers of both near-infrared cameras are summarized in Table 5 and Table 6. Small differences between the values are noticeable and the fact that the values for the multiSPEC 4c are provided with higher accuracy.

Reflectance values theoretically are represented in percentages. The authors added the decimal values which must be set in the processing software.

Reflectance *R* [%] is computed according to a formula, where *DN* stands for a digital number, and *a* and *b* are coefficients:(4)$$R=a_{i}DN+b_{i}$$

Reflectance maps were created for each band to compute vegetation indices. The pixel size for such maps depends on the ground sampling distance. All datasets were resampled to have the same pixel size of 12 cm.

## 4. Results

### 4.1. Postprocessing Enhancement Scheme

Multispectral datasets are extremely complicated to process simultaneously. The task is even harder when one wishes to generate weekly or monthly mapping over the same area. This is because a huge amount of data is created, and it must be stored according to a clear pattern so that it can be understood by other users.

The computation of multiple vegetation indicators is also a burdensome task. Unfortunately, by the time this research was conducted, none of the photogrammetric software products offered multiple computations of vegetation indices. The pix4dmapper software has one drawback in terms of saving project data. The project structure is very strict, and the individual processing steps are named in a predefined way. It is also possible to draw or import working polygon vectors into the software (only *.shp is available). To compute an NDVI, for example, one needs to choose the working polygon vector within the index that will be computed. The results are visualized and stored in a project tree. If a user chooses to compute an NDVI for a second polygon, the output raster from the first computation will be rewritten, and thus, the data will be lost. It is, of course, possible to rename the first output so that it will not be rewritten. This method is a manual process and is time consuming.

The above Is a quite relevant issue in the age of ‘big data’, especially if one deals with a great number of datasets and needs to compute numerous datasets at once. In such cases, working with pix4Dmapper might be quite unnerving.

The authors concluded that it would be necessary to use a third-party solution to prevent dealing with predefined project structures (see Figure 5). The pix4mapper software was used to apply the photogrammetric work scheme mentioned in Figure 6. The vegetation indices and other analyses had to be conducted outside of the software. The authors chose to write a script that would be capable of the automatic computing of all possible indices for the cameras, multiSPEC 4c and Sequoia, used during this project. The open-source platform Python [26] was chosen. This programming language, along with its libraries, is one of the most popular programming languages in the geospatial world. Computation took place in the QGIS Python console [26]. In order to code in QGIS, one must download the entire OSGeo4W package from the official QGIS website. Coding additional functions based on Python in the QGIS interface is very user-friendly, which enables coding and visualizing data at the same time.

Figure 6 shows the photogrammetric workflow chosen by the authors. The workflow was created after repeatedly processing and reprocessing the multispectral data. Unique in the following scheme is the fact that vegetation computing and statistical analyses were carried out in the open-source QGIS software, and an automatic algorithm was created to speed up the process of multiple VI computing, saving, clipping, and viewing of the rasters.

Working with multitemporal raster data can lead to accumulating a great amount of information. Moreover, for a basic user, it might even be impossible to visualize the proposed data, making it difficult to operate and cooperate with other people [27]. Storing big data locally prevents other team members from sharing information. One solution for bridging the gap between different expert teams is to create a spatial database for storing and archiving geospatial information. The authors decided to use a *PostgreSQL* database for this purpose, created with the extension *postgis* and *postgis_raster* using the Query tool in *pgAdmin* [28].

### 4.2. Image Data Analyses

The results from the field tests illustrate the accuracy of the aerial sensors under typical conditions. For a better understanding of what the relationship between the aerial and ground sensors is, the measurements from 29 May between the Sequoia and Trimble sensor were compared using a linear regression. This regression model is simple as it involves the variables *y* and *x*, which are related by:(5)E(y)=α+βx.

One variable is called ‘dependent’, and the other one is ‘independent’.

The output of the model indicated the relationship between the two variables in which the values of the one change at a constant rate as the other increases. This is called a trend or linear trend [29]. The trend between the field NDVI values and the ones extracted from the raster is shown in Figure 6.

The proportion of the variance for the dependent variable is presented by R-squared or R^2^. It gives us information on how much the dependent variable would be explained by the independent variable in the regression model. This number is between 0 and 1, where 1 indicates a strong correlation. In our case in May 2020, the correlation between the in situ measurements and the raster NDVI values were computed as R^2^ = 0.504, a relatively low but positive correlation (see Figure 7).

For the second flight campaign, eight specific field points were measured. It was important for the authors to evaluate how stable the field NDVI values were. This is why the eight points were measured five times on 29 July 2020. These points had more distinctive texture than their surroundings.

The values from each field measurement are summarized in a Table 7. The NDVI measurements from the GreenSeeker appeared to be stable throughout the day. Due to the similarity of the measurements, an average value for all of the NDVI measurements was used to correlate with the aerial image data.

The results from the correlation between the aerial values and the field ones measured at the eight study points are summarized in Figure 8, Figure 9 and Figure 10. The results of the correlation between the field and the raster NDVI values are summarized in Figure 11, Figure 12 and Figure 13.

Before comparing the image NDVI values with the field ones, an average of the NDVI values in Table 7 was computed. The averaged NDVI value was correlated with each raster dataset from both of the aerial sensors. Despite the low number of observations, the results prove a very high correlation—see Figure 8, Figure 9 and Figure 10.

Like the first flight campaign on 29 July 2020, the field points were measured in a grid of 2 m. These points were measured only once. Their values were compared to the raster values from all datasets from both sensors—see Figure 11, Figure 12 and Figure 13.

The results from Figure 10, Figure 11 and Figure 13 show great potential in the correlation between the field and aerial NDVI measurements. The NDVI value measures from the Sequoia showed a higher correlation coefficient in the morning on 29 July. The lowest correlation was shown in the evening, whereas the multiSPEC 4c sensor showed relatively good continuity in the correlation.

## 5. Conclusions

The following research was dedicated to a comparison of different aerial sensors with ground-measured spectral data. Moreover, the research had to improve the postprocessing of the multitemporal image data. The main objectives were to evaluate the performance of the airborne sensors throughout the day as the weather conditions change and compare the image data to in situ data. The goals of the authors were fulfilled through field tests and laboratory procedures.

The multiSPEC 4c and Sequoia multispectral cameras were compared to in situ NDVI data measured by a handheld spectrometer. The research consisted of multitemporal image capturing which led to accumulating big data. As the authors experienced technical limitations, an enhanced photogrammetric workflow was presented. The workflow is based on numerous image pre- and postprocessing attempts. The authors claim that this workflow may be used for any other aerial multispectral sensor. Using a third-party software solution enabled the research team to quickly postprocess their output rasters without any technical limitations by the state-of-the-art software. This leads to the conclusion that the authors fulfilled one of their research goals, namely, to enhance the postprocessing algorithm.

Two flight campaigns took place in 2020 on 29 May and 29 July. During the first campaign, several field points were measured in a grid of 2 m with a handheld spectrometer. The points were georeferenced with a GNSS receiver. The results from the correlation were discouraging because of the low biomass present in May.

During the second flight campaign, the authors expanded their evaluation as multiple flights were conducted on the same day but under different conditions in the morning, at lunch, and in the evening.

The authors tested the stability of the handheld spectrometer by performing several measurements. Eight study points were chosen to be measured by the spectrometer and georeferenced using a GNSS receiver. The points were measured five times on the same day (early in the morning, at noon, and in the evening). The field NDVI measurements proved to be quite stable. The average value from the measurements was correlated with the NDVI values extracted from the raster datasets produced by the Sequoia and multiSPEC 4c. The results show more than 80% correlation between the spectrometer and the sensors. The Sequoia had a higher correlation early in the morning, whereas the multiSPEC 4c had a higher correlation at noon and in the evening. Moreover, a few dozen of evenly distributed study points were measured by the same spectrometer. The in situ measurements were again correlated with the image values. The experiment indicated a very strong positive correlation. Like the previous evaluation, the Sequoia values had the highest correlation coefficients early in the morning (*R*^2^ = 0.914), declining slightly in the evening to *R*^2^ = 0.776. The multiSPEC 4c had the highest correlation to the field data in the afternoon (*R*^2^ = 0.937 and *R*^2^ = 0.881). The overall results show that all three types of sensors showed similar output NDVI values. More than 80% of the aerial and field data correlated in a positive trend.

One of the main objectives of the study was to evaluate two different airborne sensors and compare their image data to field spectrometer data. The results of the evaluation show that both sensors are reliable for multitemporal mapping over arable land. Improving the correlation between in situ and aerial measurements could be achieved by special radiometric calibration which was not the topic of this study. The authors concluded that for a small farm, a four-band sensor, such as the multiSPEC 4c, can play a significant role in crop mapping. In other words, farmers are not obliged to purchase expensive cameras without a specific need.

Even though extracting NDVI along a field is fast and cheap with a handheld spectrometer, this method had several disadvantages. The in situ data were not georeferenced in the device, and for an average data user, it could be quite difficult to create a visual representation from its measurements. GreenSeeker may be the best choice for conducting local measurements because the sensor showed it is stable, and its values highly correlate with the radiometrically corrected multispectral aerial data.

A further contribution to the research work was the expanding of the realm of knowledge for the practical analyses of infrared data. The results from the experiments apply only to the specific aerial sensors which were used, but the established workflow for postprocessing is intended to serve the geodetic and photogrammetric community more broadly, so that they can better understand the nature of image infrared data. The workflow could even be used by farm managers or agronomists for archiving their thematic maps, enabling their use in future yield predictions.

Thanks to the enhanced photogrammetric workflow combing commercial and open-source solutions, the authors saved an incredible amount of time postprocessing. The authors will continue their work on evaluating sensors in real agricultural cases and focus on improving the calibration procedure.

## Figures and Tables

**Figure 1 sensors-22-07693-f001:**
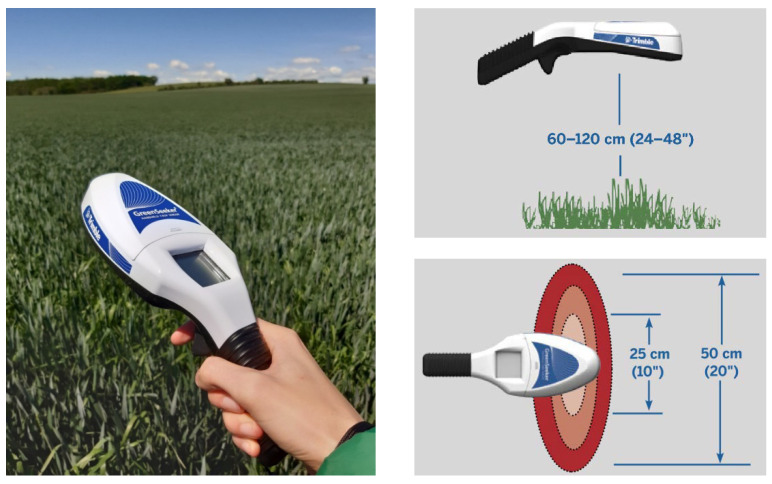
(**Left**) GreenSeeker spectrometer for ground measurements (photo: Paulina Raeva), (**right**) functionalities during NDVI field measurements.

**Figure 2 sensors-22-07693-f002:**
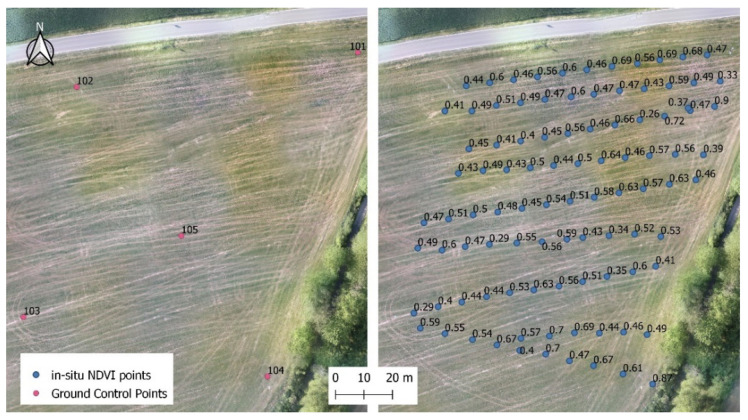
Study points in the examined field where regular in situ NDVI measurements were taken on 29 May 2020.

**Figure 3 sensors-22-07693-f003:**
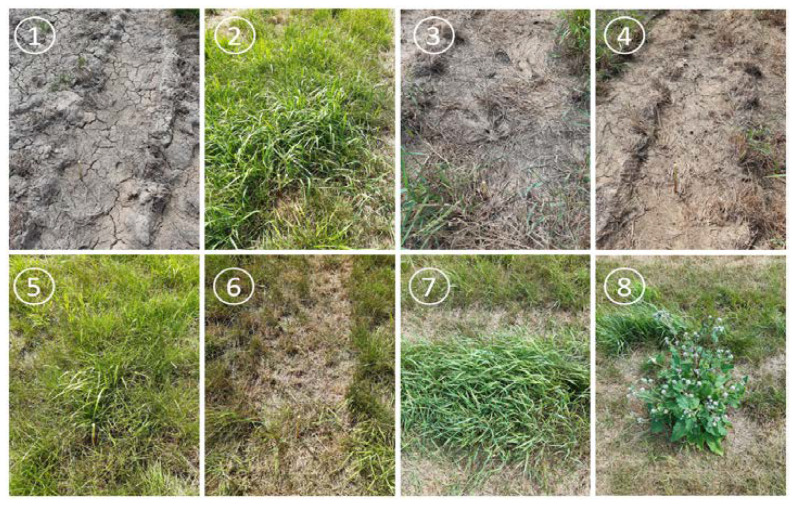
**Eight** study points in the examined field where regular in situ NDVI measurements were taken on 29 July 2020.

**Figure 4 sensors-22-07693-f004:**
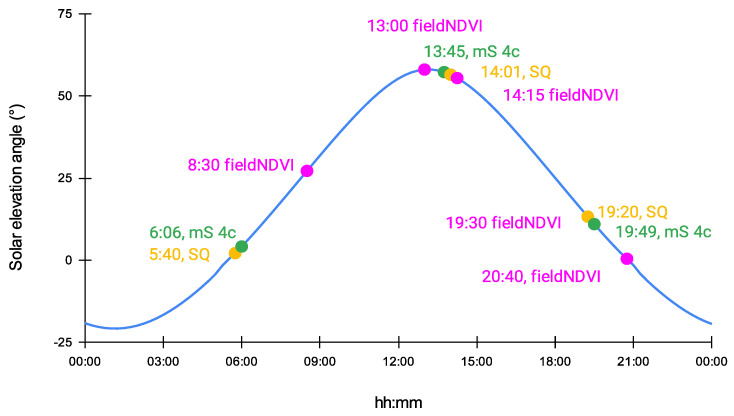
Time schedule of the aerial flights and in situ measurements mapped on a chart together with the solar elevation for that day, 29 July 2020.

**Figure 5 sensors-22-07693-f005:**
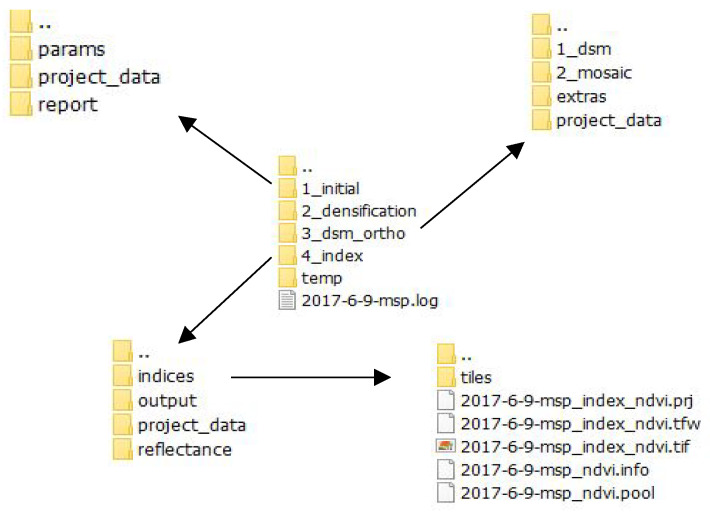
Project tree. When computing vegetation indices in pix4Dmapper, a folder of ‘indices’ is created where all outputs are stored. This is risky because all the data have the same name and, therefore, can be rewritten.

**Figure 6 sensors-22-07693-f006:**
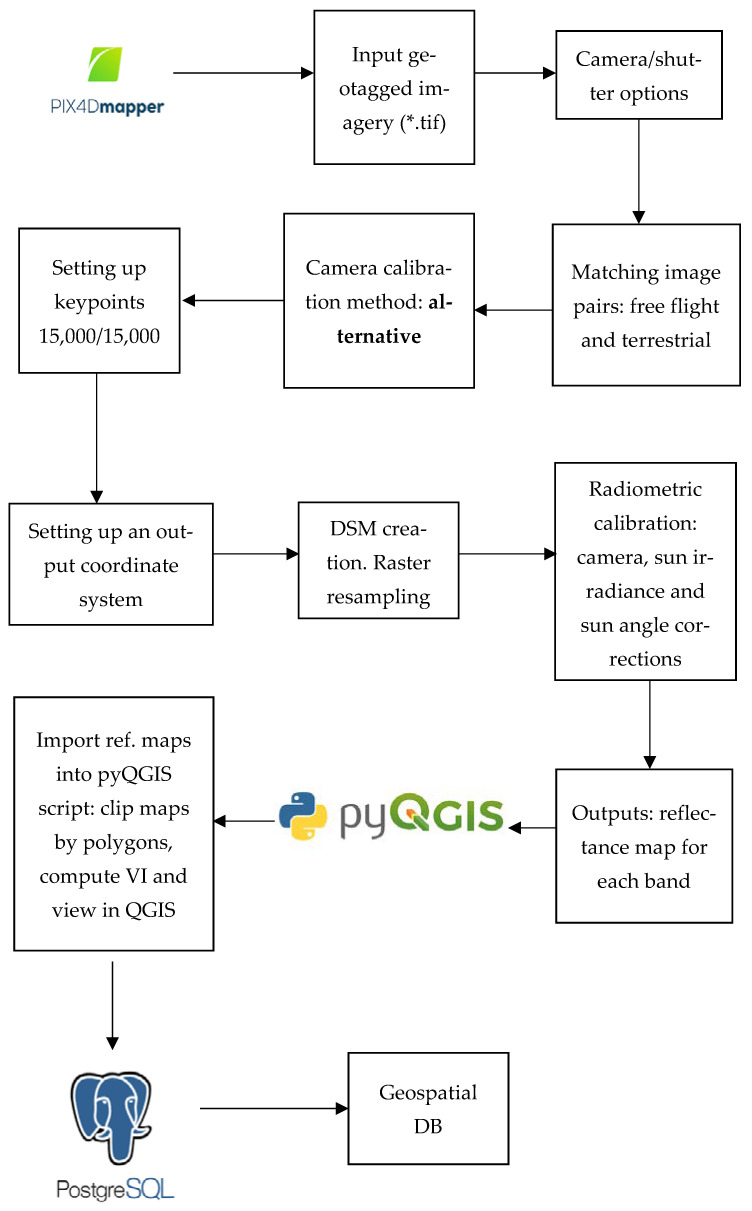
Finalized photogrammetric framework created by the authors. Pix4D served for photogrammetric processing only. Further computations were conducted in *pyQGIS*. The output rasters were added to a geospatial dataset using *PostGIS*.

**Figure 7 sensors-22-07693-f007:**
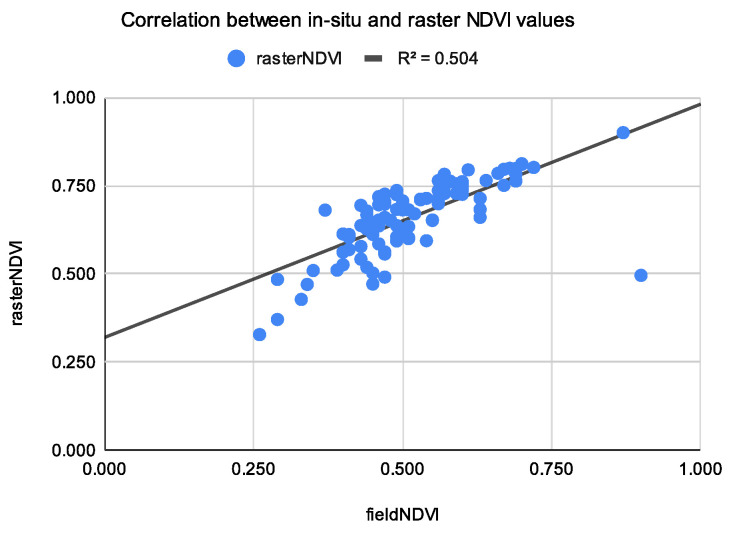
Correlation between the average NDVI values for the sample NDVI points and raster NDVI values from Sequoia, 29 May 2020.

**Figure 8 sensors-22-07693-f008:**
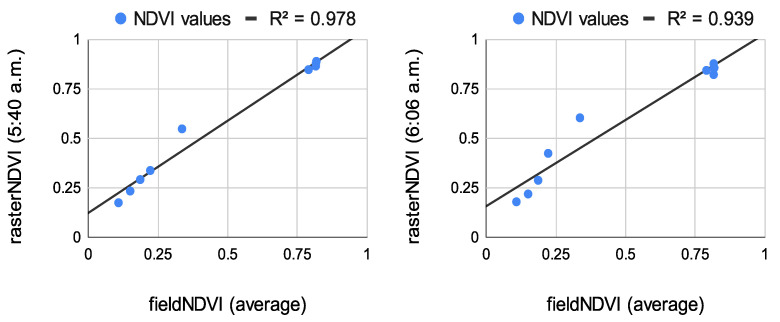
Correlation between the average NDVI values for the sample NDVI points and raster NDVI values from Sequoia (**left**) and multiSPEC 4c (**right**).

**Figure 9 sensors-22-07693-f009:**
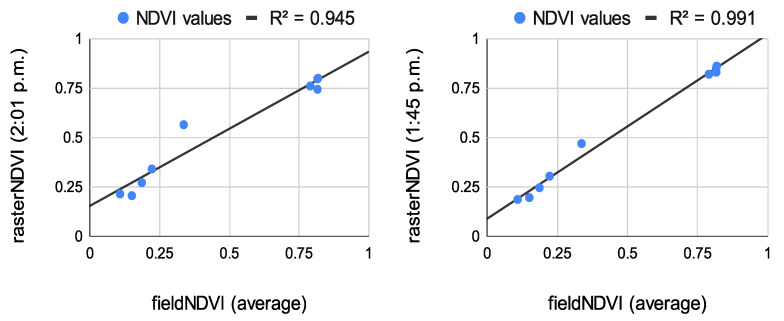
Correlation between the average NDVI values for the sample NDVI points and raster NDVI values from Sequoia (**left**) and multiSPEC 4c (**right**).

**Figure 10 sensors-22-07693-f010:**
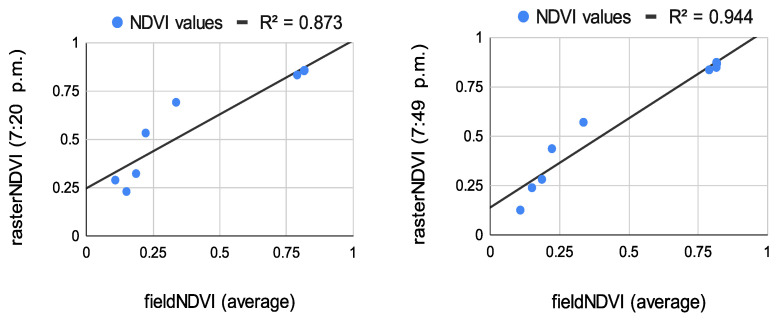
Correlation between the average NDVI values for the sample NDVI points and raster NDVI values from Sequoia (**left**) and multiSPEC 4c (**right**).

**Figure 11 sensors-22-07693-f011:**
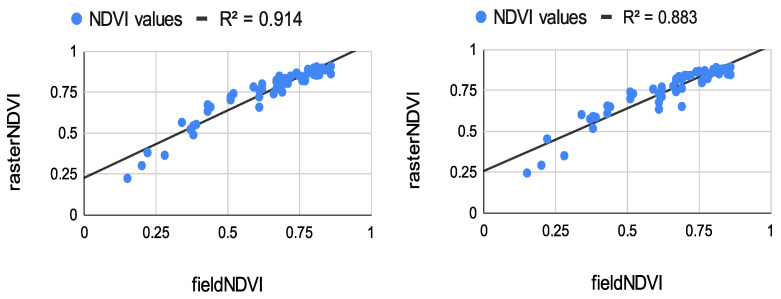
**Left**: Correlation between in situ measurements and raster values from the Sequoia sensor, flight time at 5:40 a.m. **Right**: Correlation between in situ measurements and raster values from the multiSPEC 4c sensor, flight time at 6:06 a.m.

**Figure 12 sensors-22-07693-f012:**
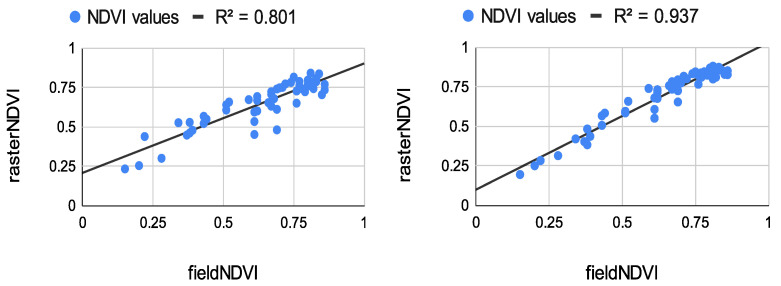
**Left**: Correlation between in situ measurements and raster values from the Sequoia sensor, flight time at 2:01 p.m. **Right**: Correlation between in situ values and raster values from the multiSPEC 4c sensor, flight time at 1:45 p.m.

**Figure 13 sensors-22-07693-f013:**
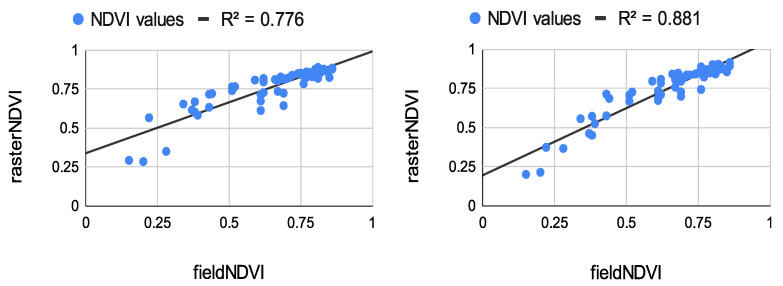
**Left:** Correlation between in situ measurements and raster values from the Sequoia sensor, flight time at 7:20 p.m. **Right:** Correlation between in situ measurement and raster values from the multiSPEC 4c sensor, flight time at 7:45 p.m.

**Table 1 sensors-22-07693-t001:** Band characteristics of the multiSPEC 4c.

Bands	Central Wavelength	Band Sensitivity	Band Width	Exposure To
Green	550 [nm]	1	40 [nm]	0.0006377
Red	660 [nm]	0.8	40 [nm]	0.0005781
Red-edge	735 [nm]	0.5	15 [nm]	0.0015666
NIR	790 [nm]	0.3	40 [nm]	0.0003865

**Table 2 sensors-22-07693-t002:** Camera and image characteristics of the multiSPEC 4c.

Aperture	Focal Length	I_W_	I_H_	Image Depth	Field of View	MP
1.8 mm	3.6 mm	1280	960	8 bit	69.5°	1.2

**Table 3 sensors-22-07693-t003:** Sequoia band characteristics, central wavelength, and bandwidth.

Bands	Central Wavelength [nm]	Bandwidth [nm]
Green	550	40
Red	660	10
Red-edge	735	10
NIR	790	40

**Table 4 sensors-22-07693-t004:** Time schedule of all photogrammetric flights conducted on 29 July 2020.

Sequoia (Start, CET)	multiSPEC 4c (Start, CET)
5:40	6:06
14:01	13:45
19:20	19:49

**Table 5 sensors-22-07693-t005:** Reflectance values for the green, red, red-edge, and near-infrared bands of the Sequoia camera.

Band	Reflectance Values [%]	Reflectance Values
Green	17.1	0.171
Red	21.3	0.213
Red-edge	26.3	0.263
NIR	36.9	0.369

**Table 6 sensors-22-07693-t006:** Reflectance values for the green, red, red-edge and near-infrared bands of the multiSPEC 4c camera.

Band	Reflectance Values [%]	Reflectance Values
Green	17.12	0.1712
Red	20.32	0.2032
Red-edge	24.68	0.2468
NIR	35.13	0.3512

**Table 7 sensors-22-07693-t007:** NDVI values measured on the study points by a handheld spectrometer at 5 different times on 29 July 2020.

No	8:30 a.m.	1:00 p.m.	2:15 p.m.	7:00 p.m.	8:40 p.m.
1	0.16	0.15	0.15	0.15	0.14
2	0.82	0.81	0.82	0.82	0.81
3	0.29	0.20	0.19	0.21	0.22
4	0.20	0.19	0.18	0.18	0.18
5	0.79	0.79	0.80	0.78	0.79
6	0.37	0.33	0.31	0.33	0.34
7	0.81	0.82	0.83	0.82	0.81
8	0.83	0.84	0.81	0.79	0.81

## Data Availability

Not applicable.

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
