# Peer review of "Evaluating the Performance of Airborne and Ground Sensors for Applications in Precision Agriculture: Enhancing the Postprocessing State-of-the-Art Algorithm"

_sensors, 2022, doi:10.3390/s22197693_

Round 1

Reviewer 1 Report

See attached file.

Author Response

Dear Reviewer,

Thank you for your review and comments on our article.

We completely agree with point 1 that the article has typographical errors. These will be corrected as the paper was sent for a revision. 

The authors also agree about point 2 that there were many specific terms in the article which can be referenced only to specific software. Using the exact terminology as state by the manufacturer is crucial for future studies which concern the same technology and software. The authors will explain better the specific terminology in the revised version.

Regarding point 3, the authors claim that it is necessary to have the reflectance values in both percentage and decimal number. In theory reflectance is presented in percentages but when one must input a reflectance value into a software, one must use another unit. This issue will be better explained in the revised version.

The authors completely agree with point 4 and 5. The text will be corrected.

Regarding point 6, the authors think that the fluctuating correlation between both aerial sensors throughout the day is due to difference in the band sensitivity, band width and sensor precision.

The authors completely agree with point 7 as the revised paper will be oriented more on the results rather than the technical part.

Moreover, we will follow the guidance from the review and improve our abstract, citations, research design, methods and overall presentation of the results and conclusion of the paper.

Reviewer 2 Report

I have gone through the manuscript entitled "Evaluating the performance of airborne and ground sensors for applications in precision agriculture. Enhancing the post-processing state-of-the algorithm". I am little confused related to the structure of the manuscript.The manuscript is not properly arranged like Abstract,Introduction, Materials and Methods, Results, Discussion, Conclusion etc. I will tell to the author to arrange your manuscript. Other issues are listed below.

English should improve by a native person. The paper suffers from a poor English structure throughout and cannot be published or reviewed properly in the current format. The manuscript requires a thorough proofread by a native person whose first language is English. The instances of the problem are numerous and this reviewer cannot individually mention them. It is the responsibility of the author(s) to present their work in an acceptable format. Unless the paper is in a reasonable format, it should not have been submitted.

2.    The novelty of the study needs to be highlighted compare to other similar studies.

3.    Discussion is weak. The discussion needs enhancement with real explanations not only agreements and disagreements. Authors should improve it by the demonstration of biochemical/physiological causes of obtained results. Instead of just justifying results, results should be interpreted, explained to appropriately elaborate inferences. Discussion seems to be poor, didn't give good explanations of the results obtained. I think that it must be really improved. Where possible please discuss potential mechanisms behind your observations. You should also expand the links with prior publications in the area, but try to be careful to not over-reach. For the latter, you should highlight potential areas of future study.

Author Response

Dear Reviewer,

Thank you for your review and comments. This will really help us improve our paper.

The authors completely agree with the need of English correction. Despite the fact, that the text had already been corrected by a native English speaker, we will send it to another one.

We will try to explain the core and potential of out study and its scientific novelty.

Th authors completely agree that the citations must be improved as well as the results and discussion of the paper.

These changes will be made in the revised version.

Reviewer 3 Report

This manuscript is ready for publication.

Author Response

Dear Reviewer,

Thank you for your comments. We appreciate it.

Round 2

Reviewer 1 Report

The manuscript is improved relative to the prior version.

Some comments:

1. Line 88 is an incomplete sentence.

2. Check general typos and grammar (e.g. Line 354 "used to correlation..." should be "used to correlate...")

Author Response

Dear Reviewer,

Thank you for your comments. I did a few languange changes in the paper and took into account your recommendations.

Reviewer 2 Report

Accepted as it stands 

Author Response

Dear Reviewer,

Thank you for your comments. I did a few language changes. They can be tracked using the tracking tool in word.